# Validation of Wireless Harness for Measuring Respiratory Rate, Heart Rate, and Body Temperature in Hospitalized Dogs

**DOI:** 10.3390/vetsci12070626

**Published:** 2025-06-29

**Authors:** Jessie Warhoe, Sydney Simpson, Benjamin Goldblatt, Kristin Zersen

**Affiliations:** Department of Clinical Sciences, College of Veterinary Medicine and Biomedical Sciences, Colorado State University, 300 W Drake Rd, Fort Collins, CO 80523, USA

**Keywords:** vital sign monitoring, wearable technology, veterinary patient care, heart rate, respiratory rate, body temperature, wireless harness, veterinary hospital

## Abstract

Monitoring vital signs such as heart rate, respiratory rate, and body temperature is important in hospitalized dogs, but can be stressful for patients and time-consuming for staff. Currently, devices with the capability to measure vital signs require wires connecting patient and device, which can be cumbersome for the patient. A wireless device that continuously records these vital signs without needing repeated handling could improve patient comfort and clinical care. This study tested the accuracy of a wireless harness device compared to standard manual methods in 19 hospitalized dogs. Heart rate, respiratory rate, and body temperature were measured manually and by the harness over the course of hospitalization. While the harness measurements of heart rate and respiratory rate generally followed the same trends as manual measurements, the agreement was not strong enough for clinical use without confirmation. Body temperature readings were less reliable, especially in dogs with longer fur. These findings suggest that while this device may help track trends in vital signs, manual checks are still recommended before making medical decisions.

## 1. Introduction

Monitoring of hospitalized veterinary patients typically involves intermittent measurement of vital parameters, including heart rate (HR), respiratory rate (RR), and body temperature. Traditionally, these measurements are obtained manually by veterinary staff through direct patient interaction. While effective, this method presents several limitations, including gaps in data collection and patient stress associated with handling [1,2]. The accuracy and consistency of such measurements can also vary between observers, particularly in high-acuity settings with large care teams [1]. Additionally, frequent manual monitoring can be time-intensive for veterinary staff and may lead to missed critical changes in patient status between scheduled evaluations [1,2,3,4].

Continuous monitoring of patient vitals using wireless technology has emerged as a promising alternative that may enhance data collection, reduce patient stress, and improve clinical decision-making [4,5,6]. In human medicine, wearable monitoring devices have been integrated into intensive care and post-operative recovery settings, allowing for real-time data acquisition and earlier detection of deterioration [7]. The veterinary field, however, faces unique challenges, including variability in patient conformation and fur length, which may impact the accuracy and feasibility of wearable monitoring devices. Additionally, patient compliance and tolerance of wearable devices can be difficult to achieve, particularly in awake, active, or stressed patients. Traditional multiparameter monitors using ECG leads or wired connections can be impractical in non-sedated or ambulatory veterinary patients, as they increase the risk of tangling or displacement [8,9].

In recent years, several wearable veterinary monitoring devices have been developed, each employing different sensing technologies. Devices such as photoplethysmography- and ECG-based harnesses aim to provide real-time HR and RR monitoring in conscious animals [8]. Other technologies include MEMS microphone-based respiratory sensors, which detect subtle chest wall movements and cardiac sounds, potentially allowing for continuous, contactless monitoring [10]. Additionally, bioinspired acoustic transducers and stretchable composite sensors have been developed to enable vital sign monitoring through fur, reducing the need for direct skin contact [11,12]. Early evaluations suggest that these devices may provide clinically useful data, but validation against gold-standard manual measurement techniques remains necessary [9].

This study evaluated a novel harness-mounted technology that has been developed to provide continuous, wireless monitoring of HR, RR, and body temperature, while also producing an ECG tracing. Once fitted onto the patient, the device allows for non-invasive, real-time data collection without the need for frequent manual interaction. However, before such technology can be widely adopted in veterinary medicine, validation against gold-standard manual measurement techniques is essential to assess accuracy and reliability.

The purpose of this study was to evaluate the accuracy of harness-measured temperature, HR, and RR measurements against manually obtained measurements in hospitalized dogs. We hypothesized that the device measurements of HR, RR, and temperature would correlate with manual measurements.

## 2. Materials and Methods

This prospective clinical study was performed at the Colorado State University Veterinary Teaching Hospital from June 2024 through October 2024. The protocol was approved by the Institutional Animal Care and Use Committee and Clinical Review Board [protocol #4051], and informed owner consent was obtained prior to enrollment. Institutional ethical guidelines were followed throughout the study. Hospitalized dogs were considered for inclusion if they had a chest circumference between 22 and 37 inches, had no evidence of thoracic pathology, and had a temperament appropriate for wearing the device throughout hospitalization while being handled for repeated HR, RR, and temperature measurements. The chest circumference criteria corresponded to the available sizes of the harness device. Dogs were excluded if they required oxygen support, had fur clipped in the sensor areas, or if the dog was unable to change positions on its own. Dogs were withdrawn from the study if they became agitated after the harness was placed, if skin irritation occurred due to the harness or its nodes, or at any point at the discretion of the primary clinician.

The device was applied according to the manufacturer’s instructions. Chest circumference was measured using a flexible tape measure, positioned around the barrel of the chest, approximately 1–2 inches behind the axilla. Once the harness was powered on, the main buckle was fastened first, followed by rotation of the harness so that the buckle was positioned under the chest. The elastic straps were then crossed under the forelimbs and across the chest, forming an “X” configuration. Electrode gel was applied to all three electrodes, and the device was left in place for at least 10 min before comparative measurements were recorded. This waiting period allowed for saturation of fur with the gel and calibration of temperature sensors.

The patient weight, breed, body condition score (BCS) on a 9-point scale, fur length, chest confirmation, chest circumference, and harness size were recorded. Fur length was defined as short if <1 inch and medium/long if >1 inch. Chest confirmation was defined as keel-shaped or deep/barrel-shaped by the study investigator placing the harness. A keel-shaped confirmation was defined as one longer from dorsum to ventrum than it was wide from each laterality, whereas a deep/barrel-shaped confirmation was one where the length from dorsum to ventrum was approximately the same as the width from each laterality. Also, presence of heart murmur, pulmonary sounds, ambulatory status, and mentation were recorded.

Paired manual and device HR, RR, and temperature measurements were recorded every 4–8 h, as ordered by the dog’s primary clinician. Three trained study personnel participated in data collection, with at least two minutes allowed between manual measurements of HR, RR, and temperature. Each parameter was measured by a single observer at each time point, with the three trained personnel alternating across shifts. Manual measurements were collected first, and then device measurements were recorded. The manual respiratory rate was collected prior to entering the kennel. The respiratory rate was determined by counting the number of visible chest excursions over 30 s and multiplying that value by two. Then the manual heart rate was collected by cardiac auscultation. The heart rate was counted for 30 s and that value was multiplied by two. A manual rectal temperature was then acquired using the same digital thermometer (Vicks SpeedRead) using a digital thermometer sheath (Medline) coated in lubricating jelly (Medline). The exact time of manual measurements was recorded. Immediately after manual measurements were collected, the device measurements were recorded from the mobile application at a time that coincided with the minute that the manual measurements were collected. The device measurements were recorded every 1 min. The clinically acceptable absolute differences were determined by two study investigators (JW and KZ) and were predefined as ±6 brpm for RR, ±15 bpm for HR, and ±0.5 °F for temperature.

Continuous data was evaluated for normality assumptions with Shapiro–Wilk testing. If the data did not meet normality, it was converted to log scale prior to analysis. The data on the association between manual and harness measurements was analyzed using linear regression analysis using the generalized estimating equation (GEE) approach to account for multiple measurements from the same dog. The agreement between both methods was evaluated using Bland–Altman analysis [13]. Differences were calculated as manual minus harness values (manual − harness); thus, positive mean differences indicate underestimation by the harness, and negative values indicate overestimation. A *p*-value of <0.05 was used to determine statistical significance. SAS V9.4 (SAS Institute Inc., Cary, NC, USA) was used to analyze data. MedCalc for Windows, version 23.0.2 (MedCalc Software, Ostend, Belgium) was used to perform Bland–Altman analysis.

## 3. Results

### 3.1. Subjects

The study population included 19 dogs, which contributed to a total of 87 paired measurements of HR, RR, and temperature. From this population, 10/19 (52.6%) were female spayed, 5/19 (26.3%) were male castrated, 3/19 (15.8%) were male intact, and 1/19 (5.3%) was female intact. Breeds included mixed breed 6/19 (31.6%), Labrador Retriever 3/19 (15.8%), German Shepherd 3/19 (15.8%), and 1 each (5.3%) of Blue Tick Hound, Bernese Mountain Dog, American Pitbull Terrier, Bichon Frise, Golden Retriever, Great Dane, and Saint Bernard. The mean age at presentation was 5.8 years (SD = 3.4). The mean body weight was 31.7 kg (SD = 12.2). The BCS was 2/9 in 1/19 (5.3%) dogs, 4/9 in 4/19 (21.1%) dogs, 5/9 in 8/19 (42.1%) dogs, 6/9 in 1/19 (5.3%) dogs, and 7/9 in 5/19 (26.3%) dogs. A keel-shaped chest confirmation was noted in 11/19 (57.9%) dogs, and a deep/barrel-shaped chest was noted in 8/19 (42.1%) dogs, with a mean chest circumference of 29.6 cm (SD = 4.0). Short fur was present in 8/19 (42.1%) dogs and medium/long fur was present in 11/19 (57.9%) dogs. Reasons for hospitalization were gastrointestinal surgery (3/19, 15.8%), splenectomy (2/19, 10.5%), fever of unknown origin (2/19, 10.5%), wound management (2/19, 10.5%), and 1/19 (5.3%) each of seizures, post-radiation care, femoral head ostectomy, genitourinary reconstruction, acute hemorrhagic diarrhea syndrome, gastric dilation, rattlesnake bite, pancreatitis, diabetes, and TPLO explant.

All (19/19, 100%) dogs had normal bronchovesicular sounds on lung auscultation. Only one dog (5.3%, 1/19) had a grade II/VI heart murmur. All dogs were able to ambulate without assistance and were noted to have either a bright and alert (12/19, 63.2%) or a quiet and alert (7/19, 36.8%) mentation. The harness sizes were small in 6/19 (31.6%) dogs, medium in 5/19 (26.3%) dogs, and large in 8/19 (42.1%) dogs. No dogs were withdrawn from the study.

### 3.2. Respiratory Rate

For each unit increase in harness RR, manual RR increased by 2.05 brpm, indicating underestimation by the harness (95% CI: 1.65 to 2.56, *p* < 0.0001) (Table 1). Limits of agreement (LoA) ranged from −52.3 to +75.0 brpm, indicating substantial variability (Figure 1). The Bland–Altman plot (Figure 1) showed a proportional bias, where the discrepancy between harness and manual RR increased at higher RR values.

Fur length and chest conformation significantly impacted the association between harness-derived and manual RR measurements (Table 1). Dogs with medium/long fur and deep-chested dogs had larger discrepancies than short-haired and keel-chested dogs, respectively (*p* < 0.0001 and *p* = 0.05; Table 1, Figure 1). Chest circumference (*p* = 0.5885) and harness size (*p* = 0.3962) did not impact the association between harness-derived and manual RR measurements (Table 1).

### 3.3. Heart Rate

For each unit increase in harness-derived HR, manual HR increased by 1.91 bpm, indicating underestimation by the harness (95% CI: 1.58 to 2.30, *p* < 0.0001) (Table 1). Limits of agreement ranged from −40.0 to +32.7 bpm with no observed proportional bias (Figure 1). The Bland–Altman plot (Figure 1) demonstrated better agreement between harness and manual HR values than RR, but variability remained present.

Fur length (*p* = 0.4819), chest conformation (*p* = 0.5784), harness size (*p* = 0.7524), and chest circumference (*p* = 0.5194) did not impact the association between harness-derived and manual HR measurements (Table 1).

### 3.4. Body Temperature

Harness-derived temperatures were not statistically different than manual readings; however for every unit increase in harness-derived temperature, manual temperature increased by 1.34 °F (95% CI: 0.93 to 1.91, *p* = 0.1136) (Table 1). Limits of agreement ranged from −2.3 to +5.2 °F. The Bland–Altman plot (Figure 1) showed tighter clustering of data points than RR and HR.

Fur length and chest conformation significantly impacted the association between harness-derived and manual RR measurements (Table 1). Dogs with medium/long fur and those with deep-chested conformation exhibited larger discrepancies in temperature measurements than short-haired or keel-chested dogs, respectively (*p* = 0.0008 and *p* = 0.047; Table 1, Figure 1). However, harness size (*p* = 0.0752) and chest circumference (*p* = 0.1133) did not impact the association between harness-derived and manual HR measurements.

## 4. Discussion

In this population of hospitalized dogs, the harness was capable of monitoring trends in heart rate (HR) and respiratory rate (RR); however, limits of agreement (LoA) for all variables exceeded predefined clinically acceptable limits. All three parameters showed positive mean differences (manual minus harness), indicating that the harness underestimated values on average.

While the mean differences between harness and manual measurements were relatively small for all parameters, the wide LoA indicate considerable variability in individual readings. This pattern suggests that the device performs well in aggregate but is not accurate across all individual measurements. In clinical terms, this means a harness-derived heart rate may be close to the manual value in one patient and off by 30 bpm in another. When the device is used in clinical practice, it would be recommended to compare harness-derived measurements to manually obtained measurements in a patient before making clinical decisions. This approach mirrors recommendations in human healthcare, where wearable monitoring systems are used to complement but not replace manual assessment, particularly when treatment decisions are being considered [3,5,7].

Respiratory rate measurements derived from the harness showed a mean difference of 2.05 bpm compared to manual, but the LoA ranged widely from −52.3 to +75.0 bpm. Proportional bias was observed, with larger discrepancies at higher RR values. Accurate manual counts of RR are more challenging at high respiratory rates or during panting, which may have contributed to the variability between the device and manual measurements, especially at higher RRs. This potential for observer-related error could have biased the LoA results independently of harness performance, making it difficult to distinguish between device error and variability inherent in manual measurement [3,4]. The difference between a RR of 100 bpm and 70 bpm as a standalone vital is negligible clinically. Given that both of the RRs indicate that the patient is tachypneic, they also both indicate that the patient should be further evaluated. Long-haired dogs and deep-chested dogs had higher discrepancies in RR than their short-haired or keel-chested counterparts. This difference may reflect how thoracic wall movement influences the device’s sensor readings or the manual interpretation of RR in dogs with differing chest shapes. Chest circumference and harness size did not impact the association between measurement techniques.

Heart rate showed better agreement than RR, with a mean difference of 1.91 bpm, and no proportional bias was detected. However, the LoA spanned from −40.0 to +32.7 bpm. This suggests that while some device-measured HRs may be very accurate, others may differ substantially from manual values in individual patients. The association between HR measurement techniques was not impacted by fur length, chest conformation, harness size, or chest circumference, indicating a more stable measurement across body types. The lack of impact from fur length suggests that HR measurement is more resistant to sensor interference. These results support the use of the harness for pulse trend monitoring, especially since HR remained consistent across coat lengths and thoracic conformations. This may be especially relevant in clinical scenarios where continuous monitoring is needed to detect changes over time, such as in patients recovering from surgery. However, the magnitude of variability still precludes its use in real-time clinical decision-making without manual confirmation. The ECG feature of the harness, which could offer added utility in arrhythmia detection, was not evaluated in this study as this would have required the placement of ECG leads which may have impacted the mobility of the dog and could have been a confounding variable in the study.

Temperature measurements showed the narrowest LoA among all parameters, ranging from –2.3 to +5.2 °F, but still exceeded the predefined ±0.5 °F threshold. The mean difference was 1.34 °F, indicating that the harness underestimated temperature values on average. Although this difference was not statistically significant (*p* = 0.1136), the mean underestimation exceeds the predefined clinically acceptable threshold and is accompanied by wide LoA. This highlights a discrepancy between statistical and clinical significance that is important for device evaluation: while the result does not meet conventional statistical thresholds, the observed bias may still influence clinical interpretation. Fur length had a significant impact on the association of harness-derived and manual temperature measurements (*p* = 0.0008), with long-haired dogs having greater discrepancies than short-haired dogs. This may reflect thermal insulation effects from dense fur or difficulty achieving adequate sensor–skin contact. Chest conformation also impacted temperature accuracy (*p* = 0.047), possibly due to differences in heat dissipation from broader thoracic surfaces. Harness size and chest circumference had no significant impact on the association between measurement techniques.

Taken together, these findings suggest that while the harness shows promise for continuous trend monitoring, particularly for heart rate, it is not suitable for completely replacing manual measurements in hospitalized dogs. When used clinically, harness-derived measurements should be confirmed with manual measurements. This interpretation aligns with systematic reviews in human medicine, which concluded that wearable monitoring systems may support early detection but are not yet substitutes for traditional methods [7]. Fur length, chest conformation, and possibly harness fit contribute to measurement discrepancies, particularly in RR and temperature readings. Long-haired and deep-chested dogs appear especially prone to underestimation. Conversely, HR was not influenced by these factors and demonstrated the most consistent performance, making it the most reliable parameter across morphologies. Device fit and stability are critical for reliable sensor output, and conformation-related variation has been shown to impact wearable sensor performance in canine posture studies [14]. These insights are consistent with previously described limitations of wearable sensors in veterinary and human patients with variable body morphology and are important for guiding future use and refinement of wearable monitoring devices [4,15].

This study has several limitations. The primary limitation is that it was not possible to take the manual and harness measurements at the exact same time, but rather they could only be guaranteed within a minute of each other. This is particularly relevant in hospitalized patients, where vital signs can change rapidly due to pain, arousal, drug effects, or disease progression [1,5,15]. In such cases, the time offset between measurements may contribute to apparent disagreement between methods, independent of device accuracy. Environmental interference (e.g., noise, movement, signal congestion, and other monitoring equipment) was not controlled for in this study and may have contributed to variability in harness-derived measurements. Wearable devices are known to be susceptible to signal artifacts introduced by patient motion, vocalization, and ambient electrical noise, especially in dynamic hospital environments [14,15]. In veterinary critical care settings, such interferences are common and can impact the consistency of physiologic monitoring, particularly in patients requiring frequent handling, housed in high-stimulation environments, or prone to repositioning and incidental contact with kennel surfaces, which may affect sensor stability—especially for temperature and respiratory rate [1]. While these variables reflect real-world clinical conditions and thus enhance generalizability, they also introduce measurement noise that can affect agreement with manual values. Manual measurement was used as the reference standard, which itself is subject to human error, which may be especially prominent in manual measurement of respiratory rates in tachypneic and panting dogs. However, manual measurements remain the accepted clinical benchmark. The relatively small sample size and single-institution design limit generalizability, although similar sample size constraints are common in both veterinary and human wearable monitoring studies [7]. Importantly, however, each dog contributed multiple paired observations, yielding 90–95 paired measurements per parameter. Prior work has shown that this number of paired observations is sufficient to estimate bias and limits of agreement with acceptable precision in Bland–Altman analysis [16]. Finally, the predefined clinically acceptable absolute differences were determined by two study investigators and may not reflect the acceptable differences by all clinicians.

In conclusion, the wearable harness evaluated in this study reliably tracked trends in HR and RR but underestimated all parameters and demonstrated wide LoA. Fur length and chest conformation impacted the association of harness-derived and manually obtained RR and temperature measurements, while HR remained consistent across fur lengths and body shapes. These findings suggest that, although wearable devices like this harness may serve as adjuncts for trend monitoring, manual confirmation should remain the standard for clinical decision-making. This approach mirrors recommendations in human healthcare, where wearable monitoring systems are used to complement but not replace manual assessment, particularly when treatment decisions are being considered [3,5,7]. If the device is in place and changes are noted in vital parameters, this will alert the veterinary care team to evaluate the patient and may lead to more rapid recognition of clinical decline. The growing adoption of wearable monitoring systems in human intensive care highlights a shared clinical momentum toward continuous, low-disruption monitoring across species, supporting further exploration of their role in veterinary practice [3,5,6,7].

## Figures and Tables

**Figure 1 vetsci-12-00626-f001:**
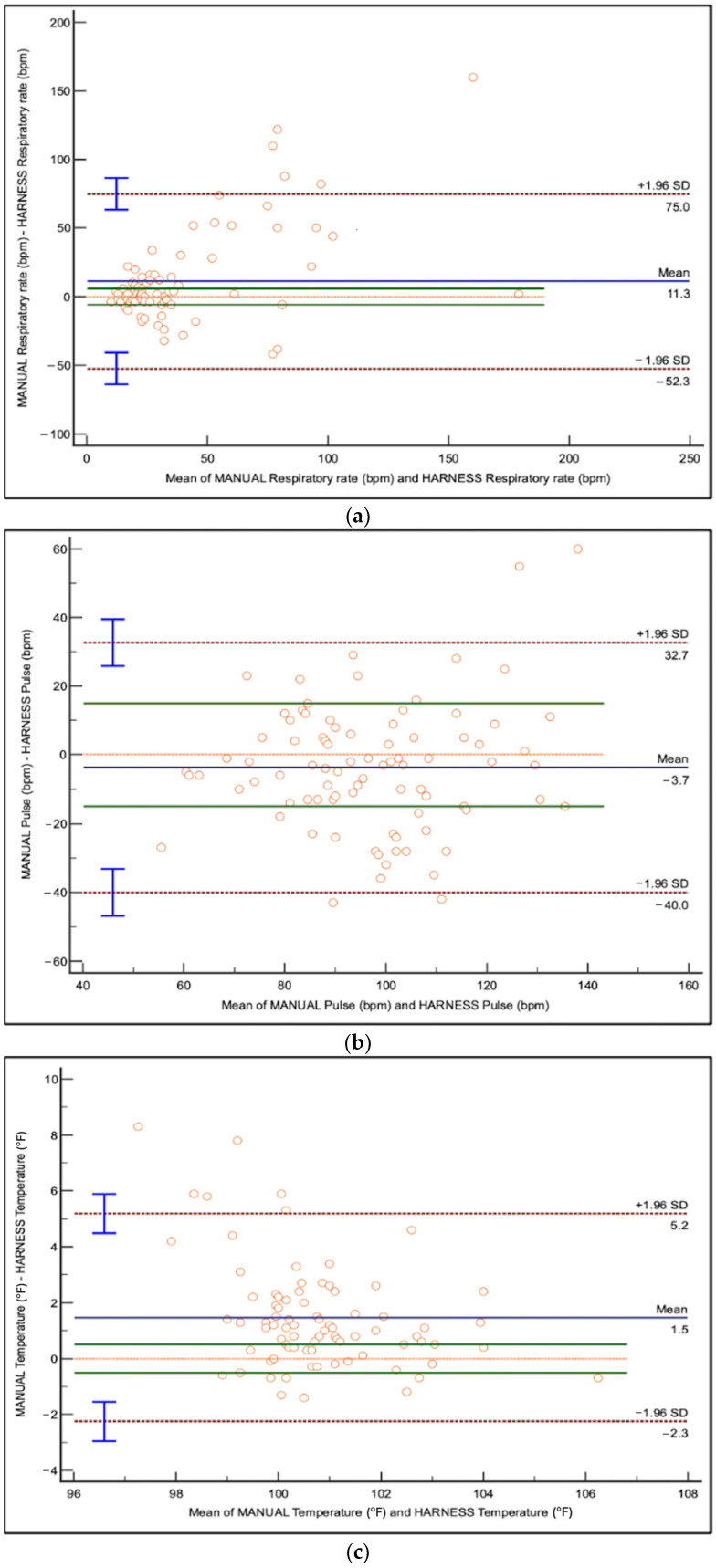
Bland–Altman plots comparing harness-derived and manually obtained measurements in hospitalized dogs: (**a**) respiratory rate (RR); (**b**) heart rate (HR); and (**c**) body temperature. In each plot, the *y*-axis represents the difference between harness and manual measurements and the *x*-axis represents the average of the two methods. The solid blue line indicates the mean difference (bias). The dashed red lines indicate the limits of agreement (LoA), calculated as the mean difference ±1.96 standard deviations. The horizontal orange dotted line represents zero difference between methods (perfect agreement). The horizontal green lines represent predefined clinically acceptable limits of absolute difference: ±6 breaths per minute for respiratory rate, ±15 beats per minute for heart rate, and ±0.5 °F for temperature. Each point corresponds to an individual paired measurement.

**Table 1 vetsci-12-00626-t001:** Agreement between harness-measured and manually measured vital signs in hospitalized dogs. Summary of mean differences, limits of agreement (LoA), and *p*-values assessing the influence of fur length, chest conformation, and chest circumference on measurement discrepancies for respiratory rate (RR), heart rate (HR), and temperature (T). Mean differences are presented with 95% confidence intervals and *p*-values for statistical significance. LoA reflect the range within which 95% of differences between harness and manual measurements are expected to fall. Bolded values indicate results that are either statistically significant (*p* < 0.05) or exceed predefined clinically acceptable thresholds: ±15 bpm for HR, ±6 brpm for RR, and ±0.5 °F for temperature.

Parameter	Mean Difference (Brpm, Bpm or °F)	Limits of Agreement	Fur Length (*p*-Value)	Chest Conformation (*p*-Value)	Chest Circumference (*p*-Value)
Respiratory Rate	+2.05 brpm (95% CI: 1.65 to 2.56, ***p* < 0.0001**)	**−52.3 to +75.0 bpm**	**<0.0001**	**0.05**	0.5885
Heart Rate	+1.91 bpm (95% CI: 1.58 to 2.30, ***p* < 0.0001**)	**−40.0 to +32.7 bpm**	0.4819	0.5784	0.5194
Temperature	+1.34 °F (95% CI: 0.93 to 1.91, *p* = 0.1136)	**−2.3 to +5.2 °F**	**0.0008**	**0.047**	0.1133

## Data Availability

Data is available through the corresponding author.

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
