# Peer review of "Validation of Wireless Harness for Measuring Respiratory Rate, Heart Rate, and Body Temperature in Hospitalized Dogs"

_vetsci, 2025, doi:10.3390/vetsci12070626_

Round 1

Reviewer 1 Report

Comments and Suggestions for Authors

The research presents data relating to the validation of wireless harness for measuring respiratory rate, heart rate, and body temperature in hospitalized dogs.

The study is very interesting and addresses a real problem in veterinary clinics because continuous monitoring of patient vitals using wireless technology is a promising alternative that may enhance data collection, reduce patient stress, and improve clinical decision-making.

The abstract is well written and summarizes the important elements of the research.

The introduction is synthetic but frames the state of the art well and clearly defines the purpose of the research.

In the section of Materials and Methods, on line 124 it is written that "Three trained study personnel participated in data collection". It is not clear whether all three assessed the physiological parameter and then the average of the 3 measurements was taken or whether they acted independently, assessing the parameter by themselves. In that case it would be necessary to evaluate the coefficient of variation between the 3 subjects.

The results are reported in a complete and detailed way.
The discussion critically deals with the data, highlighting the critical points of the research which however offers useful data to understand the reliability of the detection of physiological aprameters indirectly.
The conclusions are in line with the results.
The bibliography, although relevant, is numerically limited and should be increased.

In my opinion the paper can be accepted for publication after minor revisions

Reviewer 2 Report

Comments and Suggestions for Authors

Continuous, non-invasive monitoring is a significant interest in veterinary medicine, and this study addresses a practical clinical need.

The study has a well-defined aim to validate a wireless harness compared to manual measurements in hospitalized dogs.

The article contains several inaccuracies, such as:

Example 1: "All authors have read and agreed to the published version of the manuscript." is duplicated.

  1. Statistical Reporting:

The interpretation of p-values needs clarification:

For temperature, p = 0.1136 is reported as "not significantly different," yet the mean difference is clinically substantial (+1.34°F) with wide limits of agreement.

There is some inconsistency between statistical and clinical significance; this should be emphasized more directly.

The sample size (n = 19 dogs) is small, which limits generalizability. Although Bland-Altman analysis is appropriate, the authors should briefly discuss the power or justify the sample size.

3. Discussion Gaps:

The potential clinical impact of variability (especially the wide LoA for RR and HR) should be stressed more. Clinicians need to understand when they can trust the harness and when they should revert to manual measurement.

The ECG feature is mentioned as part of the device but is not evaluated. Even a brief note on why it was excluded would help.

  1. Limitations Section:

The authors acknowledge the limitation of time delay between manual and device measurements, but this deserves more detailed discussion since vital signs in hospitalized dogs can change rapidly.

Environmental factors and potential device interferences (e.g., kennel noise, movement artifacts) should be discussed more thoroughly.

  1. Figures and Tables:

Figure 1 Quality:

The plots are described, but the actual visuals are not provided here. Please ensure they are clearly labeled, with readable axes and legends in the final submission.

Table 1:

The table is informative but could benefit from bolding clinically significant differences or adding color coding to quickly identify where measurements exceed acceptable thresholds.

6. Discussion Gaps:

The potential clinical impact of variability (especially the wide LoA for RR and HR) should be stressed more. Clinicians need to understand when they can trust the harness and when they should revert to manual measurement.
